# Kinematic Modeling and Simulation of a New Robot for Wingbox Internal Fastening Application

**Jiefeng Jiang** [1,*] , **Jingjing You** [2] and **Yunbo Bi** [3]

1    School of Engineering, Hangzhou Normal University, Hangzhou 311121, China
2    College of Mechanical and Electronic Engineering, Nanjing Forestry University, Nanjing 210037, China
3    School of Mechanical Engineering, Zhejiang University, Hangzhou 310058, China
*    Correspondence: jiangjf@hznu.edu.cn

**Abstract:** At present, the fastener installation in a wingbox facing a narrow space must be performed manually. Using a robot is an appropriate solution for automatic assembly. However, the existing robots cannot meet the internal fastening requirements. A new robot with a prismatic joint and four revolute joints (1P4R) was developed to perform the positioning and operation in the wingbox. A compact arm link was designed, and mechanical frame structures were set up. The control system was also set up for the robot's motion. Then, the forward kinematic model was carried out with the matrix transformation method, and in the analysis the workspace entirely covered the wingbox. The inverse kinematic model was established using the geometric method, and through calculation and simulation, the inverse kinematic equations were verified and refined.

**Keywords:** wingbox; internal fastening; robot; kinematic modeling; inverse kinematics





## 1. Introduction

Assembly is a significant process of aircraft manufacturing, and its workload in the whole process accounts for 45–60%. In the aircraft assembly process, riveting and bolt joints are still the most widely used connection methods; for example, each Airbus 340 plane has 900,000 rivets and 700,000 bolts [1]. The installation of these fasteners is the most laborious part of the assembly. With the improvement of the modern assembly level, multifunction automatic riveting machines have undertaken a large number of tasks, such as hole-making, riveting, and bolt connection [2], but they generally require the working space to be open and the assembly object to be fixed. An industrial robot has been applied in aircraft drilling and riveting [3,4] as an intelligent tool with greater flexibility than automatic riveting machines.

Due to its poor openness, the wingbox is the most difficult part of the aircraft assembly. There are many bolted or riveted installation works between the skin and the main load-bearing structural parts (the spars, beams, ribs, and stringers). When the first side skin has been assembled, another side becomes more difficult because the collar or nut installation must be conducted within the wingbox's narrow space, and it must be performed manually. Because manual labor has size constraints, the efficiency is low, and there may be assembly quality instability; therefore, it urgently needs automation. Automatic fastening assembly using a robot undoubtedly is an appropriate solution. Some studies [5–9] applied a commercial 6R industrial robot to integrate the assembly systems for the fastener installation by developing the end-effector, using visual or force-sensing sensors that are mainly used for external assembly operations. Others [10–13] developed a Cartesian linear robot or SCARA robotic system for the installation of bolts and screws, which are also fit for the external fastening occasion. However, for the narrow internal assembly situation of the wingbox, existing industrial robots are obviously not applicable.

For the narrow inner working, there are some bionic robots, such as the snake robot [14]. Wright et al. [15] designed a hyper-redundant serial-linkage snake robot suited for locomotion in a pipe network [16]. OC Robotics company [17] developed a snake-arm robot system for narrow space requirements, and it implemented assembly operations in a special direction at local locations. Dong [18] designed a slender continuum robotic system for the on-wing inspection/repair of gas turbine engines. Yao [19] fabricated a prototype of a snake-arm robot with eight degrees of freedom for wingbox inner gluing, deburring, and residue removal. Zheng [20] designed a cable-driven hyper-redundant serial manipulator for internal inspection. Though the snake robot is flexible due to many freedoms, its control is complex. Its load-bearing capacity at the end is also low because of its long cantilever and flexibility. So, the snake robot is more fitted to visual inspection and simple works inside.

At present, the existing industrial robots, snake robots, cannot meet a large number of fastening assembly requirements in the wingbox, so this paper attempts to develop a new robot dedicated to internal fastening. The anthropomorphic conception was used sufficiently, and the robot design is in Section 2. The architecture of the control system and control flow are presented in Section 3. Then, in Section 4, forward kinematics are modeled using the standard D-H method, and the inverse kinematic equations are derived by using the geometric method. Finally, the workspace is analyzed using the forward kinematic model, the cases of inverse solutions are calculated and simulated, and the inverse kinematic model is verified and refined in Section 5.

## 2. Robotic Design of Mechanical System

### 2.1. Robotic Conception for Internal Fastening

The length of a wing on a large aircraft is generally more than 10 m. Here, a section of the wingbox in the wingspan direction is selected as the research model, as shown in Figure 1a. In this research, the second assembly surface on the wingbox is considered to be planar with the non-obvious curvature in the section. On the fastening plane of the wingbox model, there is an elliptical process hole (the opening) located approximately at the middle position of the Z direction. This is used for a human's arm entering the wingbox deeply to work, as shown in Figure 1b. During manual fastener installation, depending on human experience, the tool is held in the general position, and the initial contact force feedback leads to the judgment of whether the sleeve and nut align with the bolt tail. After manual adaptive adjustment, the human operates the tool finally to complete the collar or nut installation. If automation is used to replace manual assembly, an anthropomorphic robotic arm needs to be designed first that can go through the opening to enter the wingbox and reach each local assembly site under automatic control. The detailed fastening behavior would be conducted by the special fastening tool. In addition, the robot must also move along the direction of the wingspan, covering the whole wing.

The anthropomorphic concept of the robot is presented in Figure 2. There is a chassis with plane motion freedom, corresponding to human walking, which is used to determine the initial position of the robot relative to the wingbox. A prismatic pair was designed in the up-and-down direction, corresponding to human crouching and standing, to meet the requirements of the wing height direction and to drive the rear arm links to enter the wingbox through the opening. To imitate the role of a human shoulder, a rotating shaft link was innovatively designed to adapt the circumference range ($J_2$ with 360 degrees) of the assembly work by driving the rear arm's rotation around the opening. At the end of the shoulder, there is a revolute joint ($J_3$) to drive the upper arm, corresponding to the human forearm. The upper arm connects with the lower arm by a revolute joint ($J_4$), and the lower arm connects with the hand and tool using another revolute joint ($J_5$). These three joints have a parallel revolute axis, which actuates the three links swinging in some angle range, driving the tool end to reach every local fastening site. As shown in Figure 2, in principle, the linkage robot with the combination of prismatic and revolute joints can simulate various human actions for robotic end-effector positioning and internal fastening.

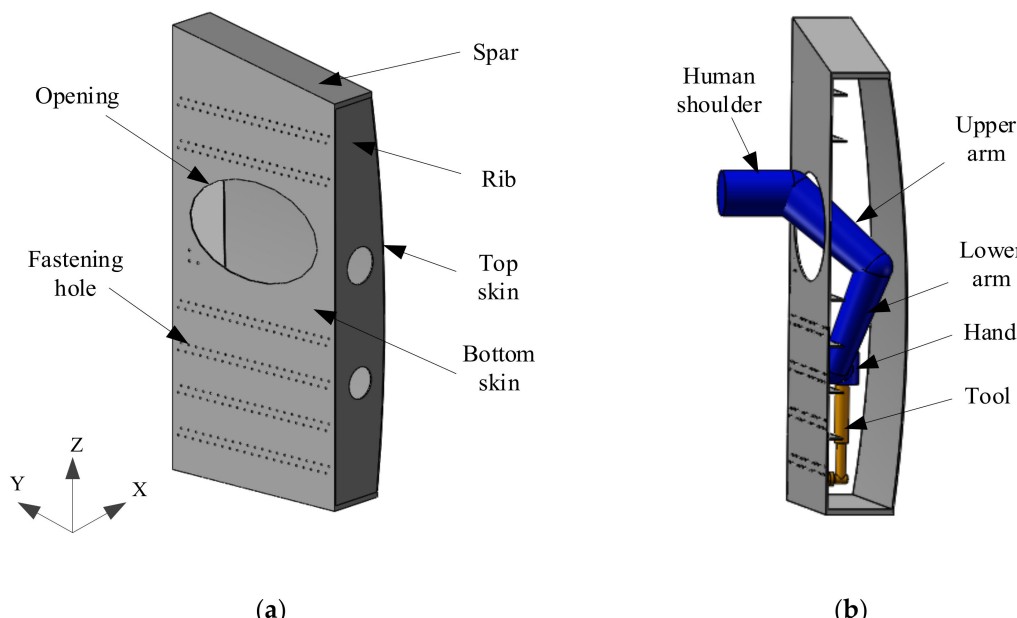

**Figure 1.** Internal fastening requirement: (**a**) Section of the wingbox model; (**b**) Manual operation using a tool (rib hidden).

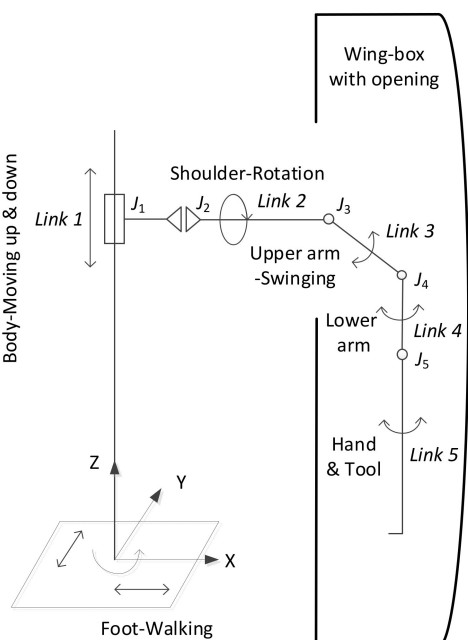

**Figure 2.** Robotic concept of anthropomorphic motion.

For the internal fastening robot, five degrees of freedom are the keys we focus on, and their link lengths need to be determined. The length of link 5 ($L_5$) is related to the length of the tool, and should be longer. For link 3 and link 4, upper and lower arms with the same length ($L_3 = L_4$) are considered. The length should be less than the thickness of the wingbox inner space to allow it to swing. Link 2 is defined by the length of the shoulder-extended part from the chassis boundary, which can be designed almost equal with link 3. $L_1$ is the linear travel distance of the prismatic joint, which plays an important role in the arm links entering the wingbox. The travel distance is related to the opening height, the length of the arm, and the tool links. Serial lengths of the links fitting the above requirements are $L_1 = 0.65$ m, $L_2 = 0.15$ m, $L_3 = L_4 = 0.11$ m, and $L_5 = 0.38$ m.

### 2.2. Robotic Mechanical Design

As the link lengths are determined, the next problem is the design of the robotic detail structures. As we know, links 3, 4, and 5 should be across the opening and work in the wingbox, and because they will suffer from narrow space constraints, the arm links should be designed to be as compact as possible. Figure 3 shows the layout of links 3, 4, and 5 in an unfolded state for presenting the arrangement relationship. The heavy lines represent the links, and the S shape of links 3 and 4 for the upper arm and lower arm are skillfully designed. The bend place of the S-shaped link can accommodate the driving motor and the corresponding transmission in two sides, not adding the dimension in the transverse direction. At revolute joint 4, links 3 and 4 overlap by the planes of the end of the S-shaped link and are arranged for the appropriate layout.

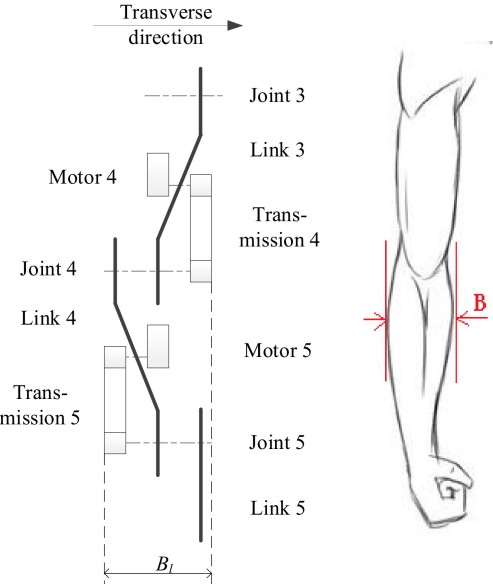

**Figure 3.** S-shaped link layout vs. human arm.

Because the initial design concept of opening size and operation space in the wingbox is fitted to human arms, the whole transverse dimension ($B_1$) of the links should be limited to less than the maximum breadth of the human arm ($B$) from the comparison in Figure 3. The S-shaped structure of the arm link is designed three-dimensionally, as shown in Figure 4. The dimension details are in [21].

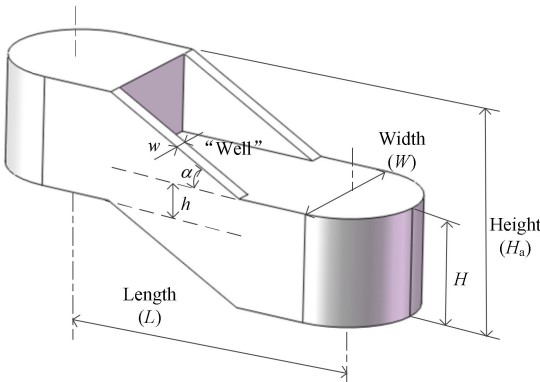

**Figure 4.** 3D structure of the S-shaped arm link.

After designing the arm links, the whole frame of the robot is considered fully as the mounting base of the links. First, the chassis of the robot manipulator is designed in the form of a wheeled cart, which is supported by three points of a universal wheel. It meets

the requirement of motion in the direction of the wingspan in order to adapt to different sections of the wing, and the wheeled cart can be used to adjust the initial position of the robot relative to the current section of the wingbox.

On the chassis, the door frame and the strengthen frame are mounted, and the slide-ways for the robot's up and down movement are installed on the back of the door frame. The chassis and the frames are welded square tubes, as shown in Figure 5a. Considering the counterweight and the whole structural stability, the door frame is placed at the rear of the chassis. At the same time, the beam is arranged as a bridge to connect the door frame and the shoulder. The shoulder (link 2) is designed as a typical mechanical shaft structure to obtain 360° rotation, and it is mounted at the end of the bridge beam. The arm links are jointed at the end of the rotating shaft, and the fastening tool is mounted on the hand. Synchronous belt transmission was chosen for five motion degrees of freedom. According to the robotic design, the prototype of the fastening robot was fabricated, as shown in Figure 5b,c.

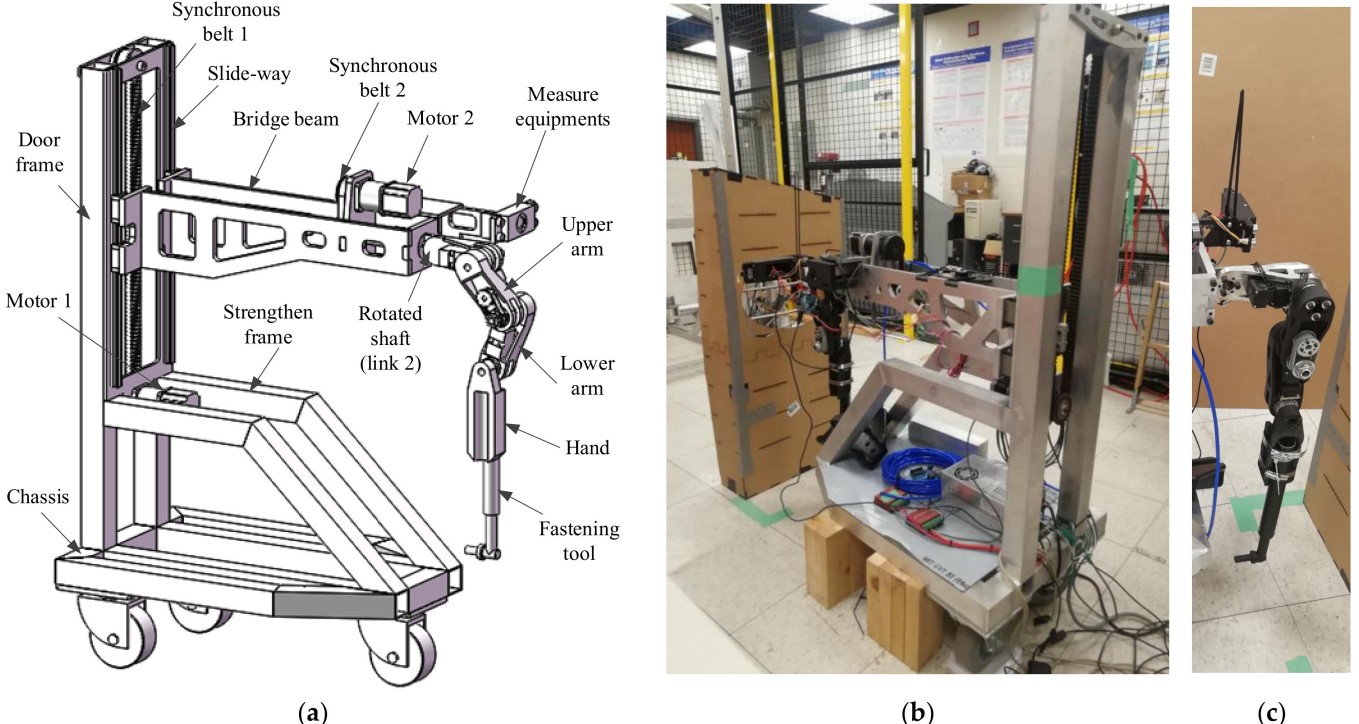

**Figure 5.** Robot for fastening in the wingbox: (**a**) Robotic frame structure; (**b**) Robotic prototype; (**c**) Robotic arm links.

## 3. Robotic Control System

The control system of the fastening robot includes two aspects: the motor control and the measurement module, the architecture of which is shown in Figure 6. At the heart of the control network is the main CPU of the system, which acts as the master unit for the robot. This takes the form of a laptop PC running Windows, which is, in turn, connected to the system of microcontrollers using a USB hub. The PC communicates via UART to the three microcontrollers for vertical prismatic joint 1, shoulder revolute joint 2, and arm joints 3 to 5. To accomplish this base-level communication, an instance of MATLAB is running on the PC, which serves as the bus controller. The measurement feedback devices consist of a monocular camera and two ultrasonic sensors. The camera is connected to the PC by a USB hub. The communication between the ultrasonic sensors and the PC requires an Arduino controller in the middle. Measurement devices are mainly used to determine the initial pose of the robot and the wingbox.

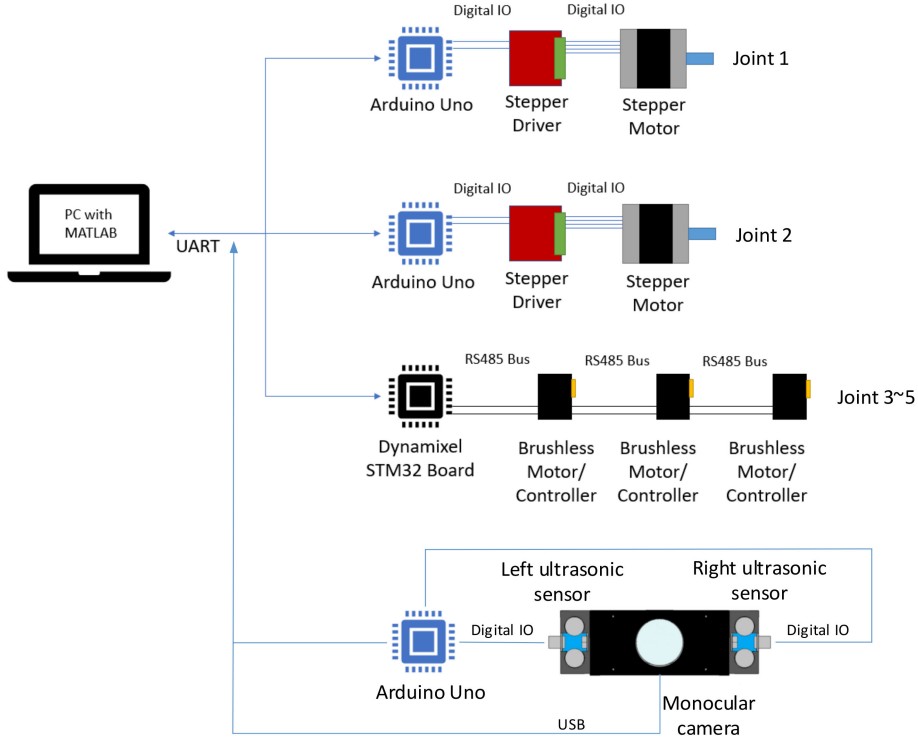

**Figure 6.** Control system architecture.

The robot control program was developed based on the MATLAB platform. The functional modules are shown in Figure 7. The program of the measurement module is used to process the images collected by the camera to measure the distance between the robot and the wingbox through the ultrasonic sensors. This provides the initial pose of the robot relative to the wingbox. Then, based on the known digital model of the robot and the wingbox, obstacle avoidance motion planning is put forward. Finally, the robotic kinematic model is established, and the corresponding control program is written. It is used for controlling the robot to enter the wingbox through the opening and move and locate the local fastening positions in the wingbox.

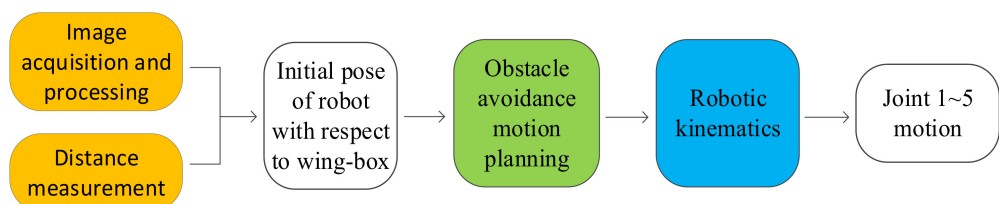

**Figure 7.** Control flow chart for robot's motion.

Robotic kinematics is the most important algorithm to realize motion control. The following section focuses on the kinematic modeling for the fastening robot. The measurement method and algorithm, along with obstacle avoidance motion planning, will be specially researched in another study work.

## 4. Kinematic Modeling

### 4.1. Forward Kinematic Model

Firstly, the valid links are extracted and simplified from the actual robot (Figure 5). In particular, the prismatic joint translating distance is regarded as virtual link 1. The beam of the robot is not a link due to having no effect in motion analysis, and link 2 is only a part of the shaft extended out of the robotic chassis. Then, the coordinate systems of the robot are

established at the joints using the D-H method, as shown in Figure 8. For the links, link $i$ is in between joint $i$ and joint $i + 1$. Regarding the coordinate frame, $Z_i$ is the rotational axis or translational axis of joint $i + 1$, axis $X_i$ generally follows the link direction or equals $Z_{i-1} \times Z_i$, and $Y_i = Z_i \times X_i$ follows the principle of the right hand. The tool end should be aligned with the hole axis in the fastening work, so link 5 is always parallel to the assembly surface of the wingbox.

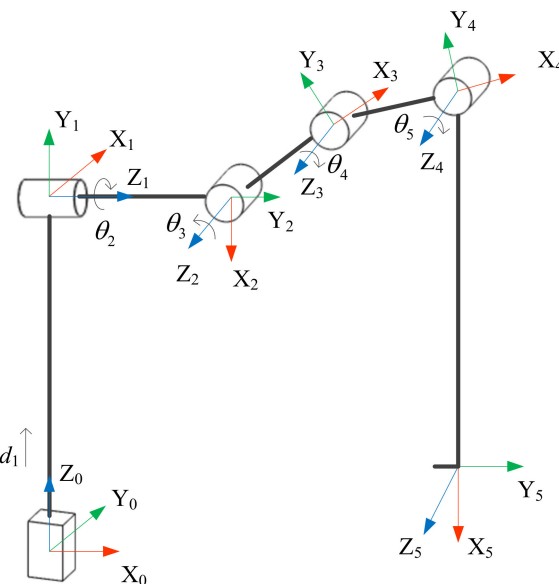

**Figure 8.** Standard D-H coordinate systems.

The parameters of links and joints obtained from standard D-H coordinate systems are listed in Table 1, where the length of link $i$, $a_i$, is the distance from $Z_{i-1}$ to $Z_i$; $\alpha_i$ is the torsional angle of the link rotated about $X_i$ from $Z_{i-1}$ to $Z_i$; $\theta_i$ is the joint angle rotated about $Z_i$ from $X_{i-1}$ to $X_i$, to be taken positive when the rotation is made counterclockwise; and $d_i$ is the joint distance from axis $X_{i-1}$ to $X_i$. For the fastening robot, the joint variable is the translated displacement for prismatic joint $d_i$ ($i = 1$), and it is the rotation angle for the revolute joint $\theta_i$ ($i = 2, 3, 4, 5$). Other parameters of the robotic D-H model are the constants.

**Table 1.** D-H parameters.

| Link | $a_i$ (m) | $\alpha_i$ (rad) | $d_i$ (m) | $\theta_i$ (rad) | Variable Scope (m or rad) |
|------|-----------|------------------|-----------|------------------|---------------------------|
| 1 | 0 | $\pi/2$ | $d_1$ | $\pi/2$ | $[0, 0.65]$ |
| 2 | 0 | $\pi/2$ | 0.15 | $\theta_2$ | $[-\pi, \pi]$ |
| 3 | 0.11 | 0 | 0 | $\theta_3$ | $[0, 3\pi/2]$ |
| 4 | 0.11 | 0 | 0 | $\theta_4$ | $[-\pi, 0]$ |
| 5 | 0.38 | 0 | 0 | $\theta_5$ | $[-\pi/2, \pi]$ |

According to the joint variable and the other known parameters of robotic links, the robotic posture of the end-effector with respect to the base coordinate frame can be calculated with direct kinematics. It can be expressed by the following coordinate transformation equation:

$$^0T_5 = {}^0T_1(d_1) \cdot {}^1T_2(\theta_2) \cdot {}^2T_3(\theta_3) \cdot {}^3T_4(\theta_4) \cdot {}^4T_5(\theta_5) \qquad (1)$$

where $^{i-1}T_i$ is the $4 \times 4$ homogeneous transformation matrix of coordinate frame $i$ relative to $i-1$, and where $^0T_5$ belongs to the special Euclidean space SE(3). By using the D-H parameters in Table 1, the detail of each pose transformation can be calculated as follows:

$$^0T_1(d_1) = \begin{pmatrix} 0 & 0 & 1 & 0 \\ 1 & 0 & 0 & 0 \\ 0 & 1 & 0 & d_1 \\ 0 & 0 & 0 & 1 \end{pmatrix} \tag{2}$$

$$^1T_2(\theta_2) = \begin{pmatrix} \cos\theta_2 & 0 & \sin\theta_2 & 0 \\ \sin\theta_2 & 0 & -\cos\theta_2 & 0 \\ 0 & 1 & 0 & 0.15 \\ 0 & 0 & 0 & 1 \end{pmatrix} \tag{3}$$

$$^{i-1}T_i(\theta_i) = \begin{pmatrix} \cos\theta_i & -\sin\theta_i & 0 & a_i\cos\theta_i \\ \sin\theta_i & \cos\theta_i & 0 & a_i\sin\theta_i \\ 0 & 0 & 1 & 0 \\ 0 & 0 & 0 & 1 \end{pmatrix}_{i=3,4,5} \tag{4}$$

Take Equations (2)–(4) into Equation (1), and the analytical solution for the posture of the end-effector can be obtained as follows:

$$^0T_5 = \begin{pmatrix} & ^0R_5 & & P_x \\ & & & P_y \\ & & & P_z \\ 0 & 0 & 0 & 1 \end{pmatrix} \tag{5}$$

where $^0R_5$ is the rotation matrix ($3 \times 3$), and the position of the end-effector can be represented as follows:

$$P_x = 0.15 + 0.11\sin\theta_3 + 0.11\sin(\theta_3 + \theta_4) + 0.38\sin(\theta_3 + \theta_4 + \theta_5) \tag{6}$$

$$P_y = 0.11\cos\theta_2\cos\theta_3 + 0.11\cos\theta_2\cos(\theta_3 + \theta_4) + 0.38\cos\theta_2\cos(\theta_3 + \theta_4 + \theta_5) \tag{7}$$

$$P_z = d_1 + 0.11\sin\theta_2\cos\theta_3 + 0.11\sin\theta_2\cos(\theta_3 + \theta_4) + 0.38\sin\theta_2\cos(\theta_3 + \theta_4 + \theta_5) \tag{8}$$

*4.2. Inverse Kinematic Model*

On the contrary, the inverse kinematic equation is

$$q_i = f^{-1}(^0T_5) \tag{9}$$

If the end-effector pose $^0T_5$ is given through inverse computation, the solution for joint variable $q_i$ is obtained. In the fastening robot, $q_i$ represents $d_1$ or $\theta_i$ ($i$ = 2, 3, 4, 5).

The fastening robot has three revolute joints, 3, 4, and 5, with a parallel Z axis, according to the Pieper criteria, and it has a closed-form solution. Because the inverse transformation method requires a large amount of computation, the geometric method was chosen here for inverse kinematic modeling, as shown in Figure 9. This assumes that the prismatic joint of the robot $\{O_1\}$ is fixed relative to the assembly object after the arm links enter the wingbox, so $d_1$ is considered a constant value ($d_1 = H_e$; $H_e$ is the height of opening center). Now, the inverse kinematic problem of the robot is to solve the joint variables $\theta_2$, $\theta_3$, $\theta_4$, and $\theta_5$.

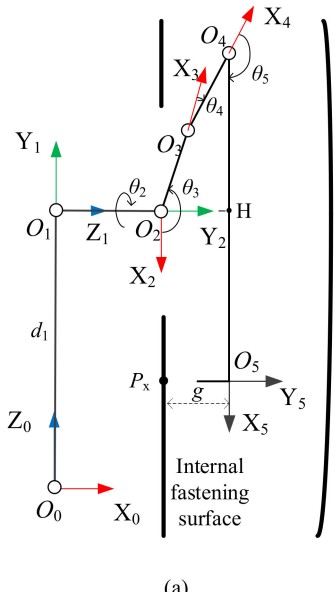
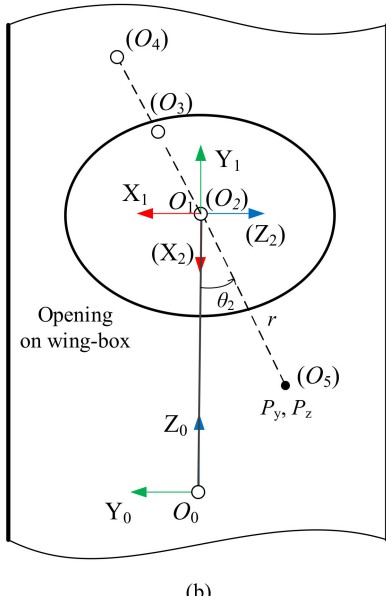

(a)                                                                                   (b)

**Figure 9.** Geometric model of robotic general posture: (**a**) Front view; (**b**) Left view.

Viewed from Figure 9, because the end-effector needs to align with the installation hole and link 5 is always parallel to the internal fastening surface in the fastener installation, the rotation matrix of coordinate frame $\{O_5\}$ with respect to $\{O_0\}$ can be obtained:

$$
{}^0R_5 = \begin{pmatrix} 0 & -1 & 0 \\ 0 & 0 & -1 \\ -1 & 0 & 0 \end{pmatrix} \tag{10}
$$

The fastening site on the internal surface of the wingbox can be presented by $P(P_x, P_y, P_z)$ in the frame $\{O_0\}$. It mentions that there is a gap ($g$) between the coordinate origin of $\{O_5\}$ and the fastening site. So, the actual position of the robotic end-effector is $O_5$ ($P_x + g$, $P_y$, $P_z$), and the homogeneous transformation matrix of the coordinate frame $\{O_5\}$ relative to $\{O_0\}$ becomes the following (from Equation (5)):

$$
{}^0T_5 = \begin{pmatrix} 0 & -1 & 0 & P_x + g \\ 0 & 0 & -1 & P_y \\ -1 & 0 & 0 & P_z \\ 0 & 0 & 0 & 1 \end{pmatrix} \tag{11}
$$

First, because $d_1$ is considered a constant value, the posture of the tool end $\{O_5\}$ relative to the coordinate system $\{O_1\}$ is expressed for the following calculation. According to Equation (1), ${}^1T_5$ can be derived by using the inverse transform method:

$$
{}^1T_5 = \left({}^0T_1\right)^{-1} \cdot {}^0T_5 \tag{12}
$$

Calculating the inverse transformation of ${}^0T_1$ in Equation (2) and using Equation (11), the detail of ${}^1T_5$ can be obtained:

$$
{}^1T_5 = \begin{pmatrix} 0 & 0 & -1 & P_y \\ -1 & 0 & 0 & P_z - H_e \\ 0 & -1 & 0 & P_x + g \\ 0 & 0 & 0 & 1 \end{pmatrix} \tag{13}
$$

The relative position of the fastening site ($^1P$ ($^1P_x$, $^1P_y$, $^1P_z$)) can be presented as follows:

$$\begin{cases} ^1P_x = P_y \\ ^1P_y = P_z - H_e \\ ^1P_z = P_x + g \end{cases} \tag{14}$$

From the left view of Figure 9b, $O_1$ coincides with $O_2$. Considering $\theta_2$, the rotation angle of the second joint, $r$ is the distance from the end-effector to the second joint shaft, and the relative position can be presented by the following:

$$\begin{cases} r \sin \theta_2 = -^1P_x \\ r \cos \theta_2 = -^1P_y \end{cases} \tag{15}$$

Combining Equation (15) with Equation (14), $\theta_2$ can be solved as follows:

$$\theta_2 = arc\tan(^1P_x/^1P_y) = arc\tan \frac{P_y}{P_z - H_e} \tag{16}$$

where $\theta_2$ can rotate in the range $[-\pi. \pi]$.

Further, we have the geometric relation equations from the front view of Figure 9a:

$$a_3 \cos \theta_3 + a_4 \cos \theta_{34} = O_4 H \tag{17}$$

$$a_3 \sin \theta_3 + a_4 \sin \theta_{34} = O_2 H \tag{18}$$

where $a_3$, $a_4$, and $a_5$ are the lengths of link 3, link 4, and link 5, respectively. The figure $\theta_{34}$ is the sum of $\theta_3$ and $\theta_4$, and $O_2 H$, $O_4 H$ can be presented in relation to the known parameters:

$$O_2 H = {}^1P_z - d_2 = P_x + g - d_2 \tag{19}$$

$$O_4 H = \sqrt{{}^1P_x{}^2 + {}^1P_y{}^2} - a_5 = \sqrt{P_y{}^2 + (P_z - H_e)^2} - a_5 \tag{20}$$

where $d_2$ is the joint distance from axis $X_1$ to $X_2$.

Squaring and summing Equations (17) and (18) yields

$$O_2 H^2 + O_4 H^2 = a_3{}^2 + a_4{}^2 + 2a_3 a_4 \cos \theta_4 \tag{21}$$

from which

$$\cos \theta_4 = \frac{O_2 H^2 + O_4 H^2 - a_3{}^2 - a_4{}^2}{2a_3 a_4} \tag{22}$$

The angle $\theta_4$ can be computed as follows:

$$\theta_4 = \pm \arccos \left( \frac{(P_x + g - d_2)^2 + \left( \sqrt{P_y{}^2 + (P_z - H_e)^2} - a_5 \right)^2 - a_3{}^2 - a_4{}^2}{2a_3 a_4} \right) \tag{23}$$

Because $\theta_4 \in [-\pi, 0]$, the value with the positive sign is ignored.

Having determined $\theta_4$, the angle $\theta_3$ can be derived as follows. Substituting $\theta_4$ into Equations (17) and (18) yields an algebraic system of two equations in the two unknowns, $\sin \theta_3$ and $\cos \theta_3$:

$$\sin \theta_3 = \frac{(P_x + g - d_2)(a_3 + a_4 \cos \theta_4) - \left( \sqrt{P_y^2 + (P_z - H_e)^2} - a_5 \right) a_4 \sin \theta_4}{a_3{}^2 + a_4{}^2 + 2a_3 a_4 \cos \theta_4} \tag{24a}$$

$$\cos \theta_3 = \frac{\left( \sqrt{P_y^2 + (P_z - H_e)^2} - a_5 \right)(a_3 + a_4 \cos \theta_4) + (P_x + g - d_2)a_4 \sin \theta_4}{a_3{}^2 + a_4{}^2 + 2a_3 a_4 \cos \theta_4} \tag{24b}$$

So, $\theta_3$ can be obtained by Equations (24a) and (24b):

$$\theta_3 = \arctan \frac{(P_x + g - d_2)(a_3 + a_4 \cos \theta_4) - (\sqrt{P_y^2 + (P_z - H_e)^2} - a_5)a_4 \sin \theta_4}{(\sqrt{P_y^2 + (P_z - H_e)^2} - a_5)(a_3 + a_4 \cos \theta_4) + (P_x + g - d_2)a_4 \sin \theta_4} \tag{25}$$

Since $\theta_3 \in [0,3\pi/2]$, the value with the positive sign is to be taken.

Because link 5 is parallel to the internal fastening surface as the installation works, the following geometric relationship of the joint angles exists:

$$\theta_3 + \theta_4 + \theta_5 = 0 \tag{26}$$

Finally, angle $\theta_5$ can be presented as follows:

$$\theta_5 = -\theta_3 - \theta_4 \tag{27}$$

## 5. Results and Analysis

### 5.1. Workspace Analysis

Using the forward kinematic model, Equations (6) to (8), and the joint variable scope in Table 1, the reachable point set of the end-effector was simulated in MATLAB 2019. The reachable workspace was obtained and is depicted in Figure 10 (blue area). It is noted that in order to reduce the amount of calculation, the large interval was chosen. If the interval is small enough, the y-z graph can be completely filled, and the outer edge boundary of the y-z graph represents the maximum reachable positions. We take the section of the wingbox into the workspace, and the black lines present the boundary of the wingbox section model, in which the max height (z) is 1.0 m, the width (y) is 0.5 m, and the max thickness (x) is 0.25 m. It can be seen that the reachable workspace of the robot completely covers this section of the wingbox. This indicates that the robotic link parameters are designed reasonably for the fastening work.



**Figure 10.** Reachable workspace of fastening robot.

### 5.2. Inverse Kinematic Calculation and Simulation Results

The analytical model of inverse kinematics was developed using the geometric method, and if the posture of the end-effector is given by Equation (11), using Equations (16), (23), (25) and (27), the joint variables ($\theta_2$ to $\theta_5$) can be calculated. However, some details, such as the sign of the joint angles, are not determined. The following is a case of inverse calculation on typical points, and, at the same time, the sign problem can be calculated and the equations can be refined.

As shown in Figure 11, $P_1$ to $P_3$ were chosen above the horizontal center line ($C_h$) of the opening on behalf of the top fastening position. $P_4$ to $P_6$ and $P_7$ to $P_9$ are distributed below the $C_h$ line, and they represent the middle and bottom assembly positions, respectively. $P_2$,

$P_5$, and $P_8$ are located on the vertical center line ($C_v$), and other points represent the further points at the boundary of the wingbox. In Equations (16), (23) and (25), set $d_1 = H_e = 0.4$ m, $g = 0.10$ m and other parameters come from Table 1, and the joint variables for typical fastening positions are calculated as listed in Table 2. At the same time, the postures of the fastening robot are simulated in MATLAB 2019 according to the calculated joint angles shown in Figure 12, where * represents the fastening point on the wingbox, and the red, purple, blue, and green lines represent links 2, 3, 4, and 5, respectively.

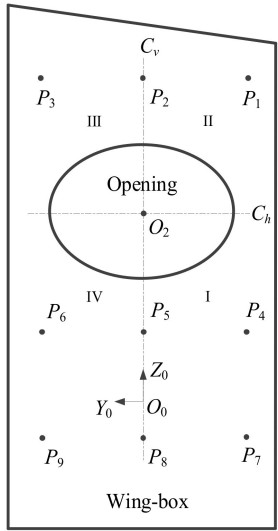

**Figure 11.** Typical fastening points on the wingbox.

**Table 2.** Joint variable calculations.

| Positions ($P_x$, $P_y$, $P_z$) (m) | $d_1$ (m) | $\theta_2$ (°) | $\theta_3$ (°) | $\theta_4$ (°) | $\theta_5$ (°) |
|---|---|---|---|---|---|
| $P_1$ (0.15, −0.20, 0.70) | 0.40 | 146.31 | 163.42 | −124.83 | −38.59 |
| $P_2$ (0.15, 0, 0.70) | 0.40 | −180 | 183.06 | −108.80 | −74.26 |
| $P_3$ (0.15, 0.20, 0.70) | 0.40 | −146.31 | 163.42 | −124.83 | −38.59 |
| $P_4$ (0.15, −0.20, 0.10) | 0.40 | 33.69 | 163.42 | −124.83 | −38.59 |
| $P_5$ (0.15, 0, 0.10) | 0.40 | 0 | 183.06 | −108.80 | −74.26 |
| $P_6$ (0.15, 0.20, 0.70) | 0.40 | −33.69 | 163.42 | −124.83 | −38.59 |
| $P_7$ (0.15, −0.20, 0.10) | 0.40 | 21.80 | 63.82 | −63.16 | −0.67 |
| $P_8$ (0.15, −0, 0.10) | 0.40 | 0 | 84.57 | −89.53 | 4.96 |
| $P_9$ (0.15, −0.20, 0.10) | 0.40 | −21.80 | 63.82 | −63.16 | −0.67 |

$P_1$ and $P_3$ are symmetric with respect to the vertical center line, and the calculated solutions are the same for $\theta_3$ to $\theta_5$, with only a difference in the sign for $\theta_2$ because it is in a different quadrant (+ in II, − in III). Similarly, $P_4$ and $P_6$ are symmetric with respect to the $C_v$ line, and there is also a difference in the sign of $\theta_2$ (+ in quadrant I, − in quadrant IV). In the value of $\theta_2$, a reasonable result is obtained by Equation (16) for $P_4$ and $P_6$. However, for $P_1$ and $P_3$, the initial solution value is the same as that for $P_4$ and $P_6$, and it is clearly irrational. The reason for this is the trigonometric function relation $\tan \theta_2 = \tan (\theta_2 \pm \pi)$. So, Equation (16) needs to be refined according to the quadrants:

$$\theta_2 = arc \tan \frac{P_y}{P_z - H_e} \qquad \text{in quadrants I and IV} \qquad (28)$$

$$\theta_2 = arc \tan \frac{P_y}{P_z - H_e} + \pi \qquad \text{in quadrant II} \qquad (29)$$

$$\theta_2 = arc \tan \frac{P_y}{P_z - H_e} - \pi \qquad \text{in quadrant III} \qquad (30)$$

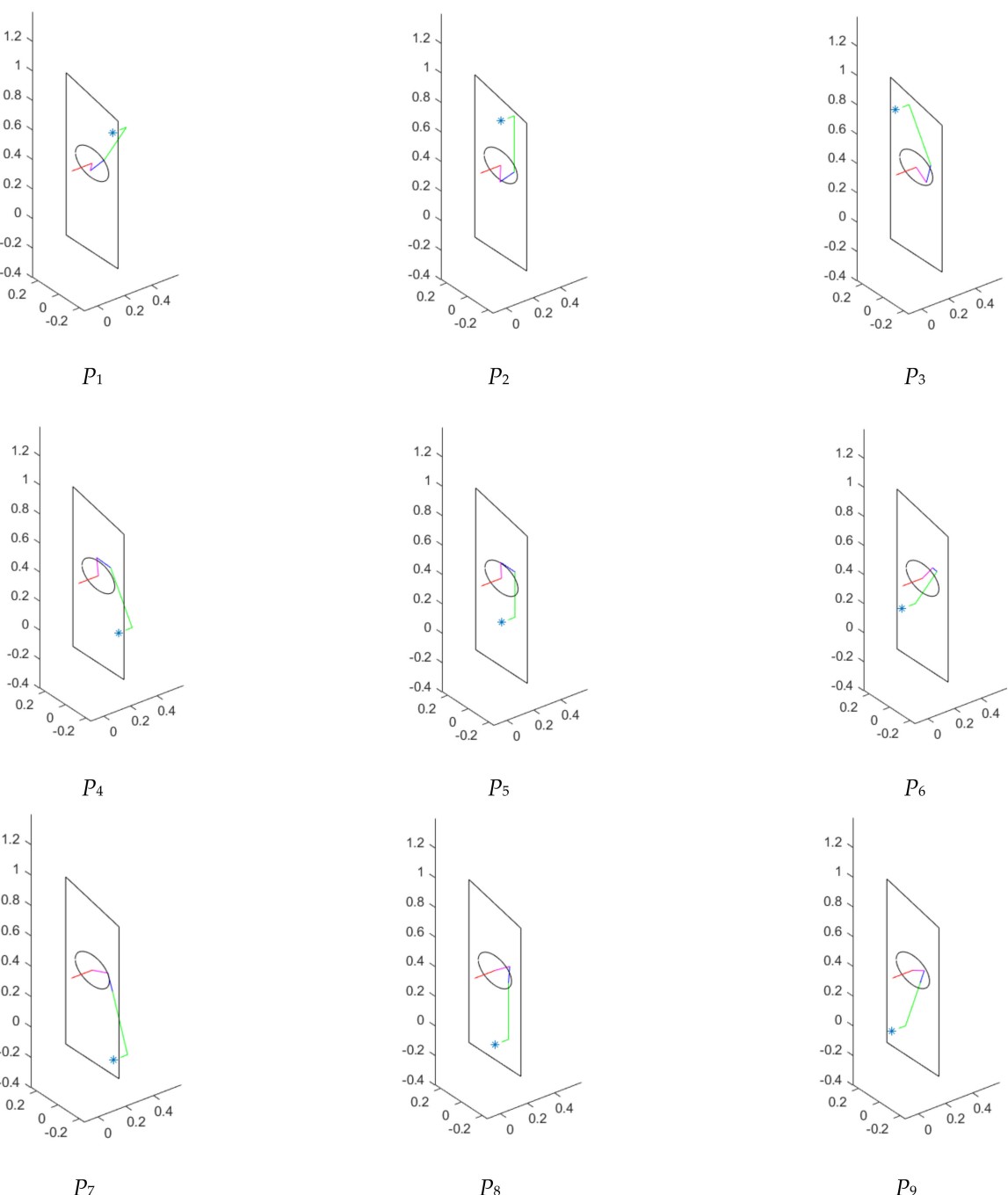

**Figure 12.** Robot poses for typical fastening points.

For $P_5$, the value and sign are correct from solving the equations, but we need to add 180° for $\theta_2$ of $P_2$. In addition, the joint angle $\theta_3$ is beyond 180° for $P_2$ and $P_5$. Because the radius of the minor axis of the elliptical opening is larger than the length of link 3, the exceeding situation is permissible, and joint angle $\theta_3$ is limited to $3\pi/2$. Compared with $P_1$ to $P_6$, $P_7$ to $P_9$ are located in the remote position, and, at this time, the calculated value of $\theta_3$ is less than 90°, $\theta_4$ is −63° or −89°, and $\theta_5$ is a small value. This indicates that all three links have the extending trend to make the end-effector reach the remote fastening point. In Figure 12, the robot poses are presented explicitly, and this confirms the calculated results. The case also indicates that the inverse kinematic model is effective.

## 6. Conclusions

Based on the process requirement of automatic fastening in the wingbox, a new robot with a prismatic joint and four revolute joints (1P4R) was put forward, and the mechanical structure was designed for the robot. The control system was also set up, and the control flow chart was proposed.

The forward kinematic model was established, and the inverse kinematic equations were derived using the geometry method. The forward kinematic simulated results indicate that the reachable workspace covers the wingbox entirely. Through the case of inverse kinematic calculation and simulation, the equation of $\theta_2$ was refined, $+\pi$ in quadrant II and $-\pi$ quadrant III. As the fastening position is far away from the opening, the joint angles $\theta_3$ and $\theta_4$ decrease to an absolute value less than $\pi/2$, and $\theta_5$ also becomes a small value.

**Author Contributions:** Conceptualization, J.J.; methodology, J.J. and J.Y.; software, J.J. and J.Y.; validation, Y.B.; formal analysis, J.J.; investigation, J.Y.; resources, Y.B.; writing—original draft preparation, J.J.; writing—review and editing, J.Y. and Y.B.; supervision, Y.B.; project administration, J.J.; funding acquisition, J.J. All authors have read and agreed to the published version of the manuscript.

**Funding:** This research is supported by Zhejiang Provincial Natural Science Foundation of China (No. LGG18E050018).

**Data Availability Statement:** All data generated or analyzed during the study are included in the article, and the data that support the findings of this study are openly available.

**Conflicts of Interest:** The authors declare no conflict of interest.

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
