# Peer review of "Kinematic Modeling and Simulation of a New Robot for Wingbox Internal Fastening Application"

_machines, doi:10.3390/machines11070753_

Round 1
Reviewer 1 Report
Comment 1: Please give the dimensions of the matrices and vectors with their sets to check the appropriateness of the mathematical equations. For example, in Equation 1, you can write $^0T^5 SE(3)$; therefore, we know that it is defined in a special Euclidean space.
Comment 2: I recommend the modification of mathematical notation throughout the paper. There are so many notation issues in the paper.
Comment 3: Lack of comparison. What is the advantage of the proposed obstacle avoidance approach?
I recommend the recheck the paper for mechanical typos. There are some grammar issues in the paper.
Author Response
Response to Reviewer 1:
Comment 1: Please give the dimensions of the matrices and vectors with their sets to check the appropriateness of the mathematical equations. For example, in Equation 1, you can write $^0T^5 SE(3)$; therefore, we know that it is defined in a special Euclidean space.
Response: Thank you for the advice. I have added the dimensions of the matrices and vectors for iTi+1, 0T5, 0R5 in line 218, 219, 223, and marked in red.
Comment 2: I recommend the modification of mathematical notation throughout the paper. There are so many notation issues in the paper.
Response: We have modified the mathematical notation in the content of the paper. Because they are large in the quantity, so we do not mark them.
Comment 3: Lack of comparison. What is the advantage of the proposed obstacle avoidance approach?
Response: Maybe I could not present clearly at the end of section 3. Actually, the main content of this paper is forward and inverse kinematic modeling and calculation, the obstacle avoidance approach will be researched and discussed in another paper.
Comments on the Quality of English Language: I recommend the recheck the paper for mechanical typos. There are some grammar issues in the paper.
Response: Thank you for the reminder, I recheck the paper for mechanical typos.
Reviewer 2 Report
Dear authors,
This is an important contribution and innovative research, good work.
The paper needs some revisions:
-The Results section should be added.
-The conclusion must include numerical data that reflects the most important results.
-Review the English wording.
-It is suggested to include more pictures of the machine.
Additional comments:
1. The authors present a research about a novel robot design for fastener installation in the wing-box.
2. Does it address a specific gap in the field? The topic is interesting, it contributes to the research line of robot kinematics.
3. The research introduces a new robot for this particular application.
4. The materials and methods sections should be added to the main structure of the article.
5. The conclusions must include numerical data that reflects the most important results.
6. Are the references appropriate? Yes
7. Please include any additional comments on the tables and figures. It is suggested to include more pictures of the machine.
8. The Results section should be added to the main structure of the article.
9. The title is simple, it should be improved and more detailed.
10. The acceptance of the article is recommended with some revisions/editions.
11. Review the English wording.
In order to improve the quality of English, it is recommended to check that the paper should be written in Third Person.
Author Response
Response to Reviewer 2:
-The Results section should be added.
Response: The original section 4 is divided to new section 4 and section 5, the new section 4 presents the method of forward and inverse kinematic modeling, and new section 5 mainly presents the results of the kinematic calculation and simulation.
-The conclusion must include numerical data that reflects the most important results.
Response: I have added the important numerical data in the conclusions, marked in red.
-Review the English wording.
Response: We have rechecked the paper for mechanical typos and the problem of Third Person.
-It is suggested to include more pictures of the machine.
Response: Thank you for the advice. I added the picture, Figure 5 (c) Robotic arm links.
Additional comments:
4. The materials and methods sections should be added to the main structure of the article.
Response: The original section 4 is divided to new section 4 and section 5, the new section 4 presents the method of forward and inverse kinematic modeling, and new section 5 mainly presents the results of the kinematic calculation and simulation.
9. The title is simple, it should be improved and more detailed.
Response: The title is modified as “Kinematic Modeling and Simulation of a New Robot for Wing-box Internal Fastening Application”, marked in red.